# Evaluation of Food-Intake Behavior in a Healthy Population: Personalized vs. One-Size-Fits-All

**DOI:** 10.3390/nu12092819

**Published:** 2020-09-15

**Authors:** Femke P. M. Hoevenaars, Charlotte M. M. Berendsen, Wilrike J. Pasman, Tim J. van den Broek, Emmanuel Barrat, Iris M. de Hoogh, Suzan Wopereis

**Affiliations:** 1TNO, Netherlands Organization for Applied Scientific Research, Research Group Microbiology & Systems Biology, P.O. Box 360, 3700 AJ Zeist, The Netherlands; femke.hoevenaars@tno.nl (F.P.M.H.); charlotte_berendsen@live.nl (C.M.M.B.); wilrike.pasman@tno.nl (W.J.P.); tim.vandenbroek@tno.nl (T.J.v.d.B.); iris.dehoogh@tno.nl (I.M.d.H.); 2Laboratoire Lescuyer, Department of Research, 15 rue le Corbusier, CEDEX, F-17442 Aytré, France; emmanuel.barrat@laboratoire-lescuyer.com

**Keywords:** personalized nutrition, food-intake behavior, phenotype, automated advice, metabolic health, randomized controlled trial, healthy subjects

## Abstract

In public health initiatives, generic nutrition advice (GNA) from national guidelines has a limited effect on food-intake improvement. Personalized nutrition advice (PNA) may enable dietary behavior change. A monocentric, randomized, parallel, controlled clinical trial was performed in males (*n* = 55) and females (*n* = 100) aged 25 to 70 years. Participants were allocated to control, GNA or PNA groups. The PNA group consisted of automatically generated dietary advice based on personal metabolic health parameters, dietary intake, anthropometric and hemodynamic measures, gender and age. Participants who received PNA (*n* = 51) improved their nutritional intake status for fruits *P* (*p* < 0.0001), whole grains (*p* = 0.008), unsalted nuts (*p* < 0.0001), fish (*p* = 0.0003), sugar-sweetened beverages (*p* = 0.005), added salt (*p* = 0.003) and less unhealthy choices (*p* = 0.002), whereas no improvements were observed in the control and GNA group. PNA participants were encouraged to set a goal for one or multiple food categories. Goal-setting led to greater improvement of food categories within the PNA group including; unsalted nuts (*p* < 0.0001), fruits (*p* = 0.0001), whole grains (*p* = 0.005), fish (*p* = 0.0001), dairy (*p* = 0.007), vegetables (*p* = 0.01) and unhealthy choices (*p* = 0.02). In a healthy population, participants receiving PNA changed their food-intake behavior more favorably than participants receiving GNA or no advice. When personal goals were set, nutritional behavior was more prone to change.

## 1. Introduction

Chronic disease prevalence is rising, and as a result we are faced with increasing health care costs [1,2]. Poor dietary intake is a major risk factor for chronic diseases [3]. A diet low in added sugars, saturated fat, sodium and alcohol, while high in fruit, vegetables, whole grains and oily fish has been acknowledged for its positive effects in preventing chronic diseases [4,5,6]. Changing everyday nutritional choices to meet dietary guidelines is one of the major leading modifiable actions in prevention of chronic noncommunicable diseases [7,8]. For example, Current eating patterns in the USA and Europe, often do not align with dietary guidelines [9,10]. This emphasizes the need for changing food-intake behavior. Personalized nutrition advice (PNA) has been shown to be an effective method to support dietary behavior change [11,12,13,14,15,16].

Personalized nutrition can be defined as the use of information specific to each individual, based on scientific evidence, to promote changes in eating behavior that can lead to measurable health benefits [17]. PNA can be based upon demographics, genetics, phenotypic characteristics, psycho-social characteristics and lifestyle behaviors of a person [15,18]. All these parameters are key characteristics of an individual and can be used to generate—after integrated analysis—specific dietary advice. Therefore, a great advantage of personalized nutrition is that individuals receive advice that is tailored to their specific needs to promote dietary behavior change. It is well known that this change in behavior is often difficult to accomplish, since new dietary behavior intentions conflict with old habits (i.e., unhealthy eating) [19,20]. Personalized support and advice may be useful to overcome this hurdle of behavior change. Previous studies already indicate that PNA can lead to healthier dietary choices compared to generic, one size fits all, nutritional advice [21,22]. It is assumed that this effect is caused by increased attention and motivation of an individual when a message is made personal [11]. In addition, personalized dietary advice is more likely to be read, perceived as relevant and remembered by an individual compared to generic advice [23].

The form in which personalized advice is given is also of importance in changing individual behavior. Research has shown that setting clear and achievable goals is effective in supporting behavior changes [24]. These personal goals for changing food-intake behavior can be translated into actions using implementation and planning techniques tailored to a person’s needs. Ability and motivation to bring goals into practice can be increased in this way. Thus, goal-setting can be an effective tool for maintaining a healthy lifestyle.

Previous studies have shown that dietary-advice systems tailored to a person’s health status are more effective in changing health parameters than a generic, one-size-fits-all, approach [25,26]. To the best of our knowledge, limited studies have implemented personalized advice systems that combine personalized advice based on health status and food-intake behavior data with behavior change techniques [22,27]. Therefore, we are interested in investigating whether the combination of PNA based on individual health status and food-intake behavior combined with a behavior change technique can be more effective in improving food-intake behavior in a healthy population than no advice or generic advice. We focus in our automated PNA on how to implement evidence-based nutritional advice, which has shown to be preventive against chronic disease outcomes, by practical tips for change. In the PNA group, goal-setting was evaluated to investigate if goal-setting for a specific food category improves food-intake behavior for that specific food category more than without a goal-setting strategy.

## 2. Materials and Methods

### 2.1. Ethics Statement

Before entering the study, all participants provided written informed consent. The study was approved by the French (medical) Ethics Committee under the study number ID-RCB Number: 2018-A00375-50, Comité de Protection des Personnes (CPP) and agreed upon by the French health agency (ANSM, Competent Regulatory Authority).The study was conducted in accordance with the Declaration of Helsinki as revised in 1983 and was registered in the Dutch Trial Register NL7054 (former ID: NTR7259).

### 2.2. Study Participants

Males and females, who considered themselves as healthy, aged 25–70 years old with a body mass index (BMI) between 20–35 kg/m^2^ were recruited from a clinical center database (Biofortis, France). Participants were unrestrained eaters, based on the three-factor eating behavior questionnaire (TFEQ, restraint scores of <13 for Factor 1 were included [28]). Those who had been prescribed medication for cholesterol, glucose, insulin, blood pressure, body weight and gastrointestinal functioning were excluded. Participants with a history of medical or surgical events that could significantly affect the study outcomes—including more than five years of cardiovascular disease or hypertension or more than two types of medication for cardiovascular disease or hypertension—were also excluded. In addition, participants who smoked regularly, used dietary supplements, suffered from chronic diseases, food allergies or intolerances were excluded from the study.

### 2.3. Study Design

The study was a monocentric, parallel, open-labeled (subjects were only informed that they were assigned to an intervention group without specification), randomized controlled clinical trial (Figure 1). After a nine-week run-in period in which participants kept their habitual lifestyle (diet and physical activity), participants were stratified to the different interventions based on different plasma parameters in response to a mixed meal drink (measured at baseline (Visit 1, V1)), gender, age and BMI. Randomization occurred via a computer-based blocked randomization table to either the free-living condition (control) group, generic nutritional advice group (GNA group) or the personalized nutrition advice group (PNA group) [29]. All staff involved in analyses of outcome variables were blinded to the intervention allocation. All participants were tested during three clinical visits, V1 at baseline (t = 0 weeks), V2 after the run-in period (t = 9 weeks) and V3 after the intervention period (t = 18 weeks). The intervention period between V2 and V3 took at least nine weeks and was expected to be sufficient to evaluate the initiation of behavior change [30]. The V3 visit was delayed for some subjects due to logistical issues. Some subjects, therefore, had a slightly prolonged intervention period (0.55 ± 1.25 weeks). The prolongation of the intervention period was not identified as a confounder and therefore not included in statistical analyses. Sample size was calculated based on previous literature which showed an effect on physiological functioning between a personalized and generic advice group [27]. With a power of 80% and Type 1 error rate of 5% an effect size of 0.36 was expected with N = 55 for physiological functioning.

### 2.4. Study Procedures

For all three visits (V1, V2, V3) participants came in to the study center (Biofortis, France) in the morning after at least 10 h fasting. During each visit, anthropometric, hemodynamic, and fasting blood parameters were measured once. At the start of the study (V1), blood parameters were measured in response to a mixed meal drink (PhenFlex test drink, PFT) for all participants. The PFT drink contained 320 mL water, 60 g palm oleine, 75 g dextrose, 20 g Protifar (protein supplement) and 0.5 g artificial aroma (total energy content 920 kcal), as described previously [31,32,33,34]. Participants were instructed to consume the test drink within 5–10 min and were not allowed to eat or drink until the last blood sampling, except from drinking water. Plasma samples were taken before (t = 0, under overnight fasting conditions) and four time-points (t = 0.5 h, t = 1 h, t = 2 h and t = 4 h) after consumption of the test drink. At V2 and V3 only plasma samples under overnight fasting conditions were used for the present study.

### 2.5. Anthropometric Parameters

Anthropometrics were measured at V1, V2 and V3, except height which was only assessed at baseline. Body weight was recorded with a calibrated weighing scale (KERN PCB 6000–1 to the nearest 0.1 kg). The measures have been performed by trained personnel of the contract research organization Biofortis Mérieux. Measures were taken once. Participants did not wear shoes or heavy clothing. Waist circumference was determined directly over the skin at the midpoint between the lower part of the last rib and the top of the hip.

### 2.6. Metabolic Parameters

Blood samples were collected in ice-cold ethylenediaminetetraacetic acid (K_2_EDTA) or Li-heparin tubes. Plasma and serum samples were stored at ≤ −20 °C for clinical chemistry and ≤ −70 °C for all other parameters after centrifugation (for 15 min at approximately 2000× *g* at approximately 4 °C within 30 min after collection). The following parameters were measured in plasma: glucose, insulin, total cholesterol, HDL cholesterol, LDL cholesterol, non-esterified fatty acids (NEFA) and triglycerides (Cobas Integra 400+ and Roche Diagnostics kits, Basel, Switzerland).

### 2.7. Insulin Resistance Indices

We calculated the homeostatic model assessment of insulin resistance (HOMA-IR) according to Song et al. [34].

### 2.8. Hemodynamic Parameters

Systolic blood pressure (SBP), diastolic blood pressure (DBP) and heart rate were measured twice after 5 min rest in sitting position (CARESCAPE V100, GE Healthcare). The average of these measurements was used for the analysis.

### 2.9. Dietary Intake and Physical Activity

Dietary intake was assessed by an adapted validated French food frequency questionnaire (FFQ), which consisted of 52 questions representing different food categories to indicate usual frequency of consumption, as well as usual portion size [35]. The participants indicated their food consumption over the last month and could report their frequencies of consumption on a daily, weekly or monthly basis. Combining frequencies and usual portion size was used to estimate daily intakes of each food categories. Physical activity was assessed by the International Physical Activity Questionnaire (IPAQ) [36]. The participants filled out the same FFQ and IPAQ at each visit (V1, V2 and V3).

### 2.10. Treatment of the Control and GNA Group

The participants in the control received no dietary advice and maintained habitual lifestyles. The participants in the GNA group received a leaflet with the general French dietary guidelines at V2. This leaflet contained guidelines on eight food categories, details can be found in Appendix A, https://www.anses.fr/en/system/files/NUT2012SA0103Ra-1EN.pdf.

### 2.11. Automated Health Feedback for the PNA Group

Automated feedback on health parameters (e.g., waist, BMI, blood pressure, glucose) and food-intake behavior was provided to the PNA group via an online personal web-portal at V2. Individual health data, e.g., body weight, blood pressure, lipids, were also integrated into a health score based on the health space model. This model was adapted from the original principle [18]. In short, this health space model was based on phenotypic flexibility from two reference groups; healthy young subjects (20–29 years) with a low to normal fat percentage (≤20% for men; ≤30% for women) and elderly (60–70 years) with a normal to high fat percentage (>20% for men; >30% for women) [31]. The individuals in our study were fitted by a net linear regression model trained on the aforementioned reference groups based on their integrated mixed meal response at baseline. A rank-based score was provided which informed participants and researchers of the health status of the participants on a metabolic age scale (≥18 years): the younger, the healthier.

### 2.12. Personalization of Dietary Advice for the PNA Group

Personalization of the dietary advice for the PNA group was based on metabolic measurements at baseline, the actual food consumed based on the FFQ, anthropometric and hemodynamic measures, as well as gender and age. The personalization of the advice was founded on previous research from the Food4Me consortium [15]. An automated PNA system implemented these results into advice. This system evaluated whether the nutrition intake status (NIS) of the food categories filled out in the FFQ were low (<50%), average (≥50–75%), above average (≥75%), optimal (intake according to guideline) or excessive (>100% in case of dairy) than nutrition guidelines. This evaluation focused on compliance with the French dietary guidelines (Programme national nutrition santé (PNNS)) combined with Dutch dietary guidelines (http://www.voedingscentrum.nl) to make some food category advice quantitative rather than qualitative. Second, the anthropometric measures and metabolic responses which were considered relevant or abnormal were used and combined with the dietary intake data into personalized dietary advice. Moreover, participants were encouraged to set a nutrition goal by formulating implementation intentions, in consultation with the dietician, for one or multiple food categories. All subjects had three contact moments with the dietician by phone: at the beginning of the PNA intervention period, after three weeks and after six weeks. The PNA group completed the FFQ just before each phone call, so in the phone call further personalization and adaptation of the food intake could be discussed. These in-between FFQs were not used for evaluation of the PNA in the present analysis. The personalized advice and implementation intentions were available on an online personal web-portal.

### 2.13. Statistical Analysis

Data were analyzed on a per protocol basis, using R statistical software, version 3.5.1. Due to the use of self-reported data, data were checked for misreports by use of the interquartile range rule. If self-reported data exceeded 3 times the SD in combination with abnormal eating, e.g., eating 30 kg of fruits per day, data points were considered as misreports and removed from analysis. Misreports were observed in all treatment groups for the food categories fruits, whole grain, spreads and oils and sugar-sweetened beverages (SSB). Ordinal mixed effect models were used to assess the following:


Changes in nutritional intake status between intervention groups:


To assess the effect of PNA on food-intake behavior, we determined changes in NIS of the PNA group between V2 and V3 and compared these to the changes in the control and GNA group. In this model, ‘time’ and ‘treatment’ (and their interaction) as well as ‘gender’ and ‘age’ were specified as fixed effects and ‘participant’ as random factor.


Goal-setting within the PNA group:


Nearly all participants in the PNA group set goals for multiple food categories. Only one participant did not set any goals and was excluded from the analysis to determine if goal-setting was effective (*n* = 50). Only food categories which were selected for goal-setting by eight or more participants were used in the analysis. To assess if goal-setting was effective, changes in NIS for goal-setting participants were compared to changes in NIS for non-goal-setters from the PNA group for that specific food category. In this model, ‘time’ (V2 compared with V1 and V3), ‘goal’ (yes or no) and their interaction, as well as ‘gender’ and ‘age’ were specified as fixed effects and ‘participant’ as random factor. For all ordinal mixed models, post hoc analyses were performed to further identify the observed time effects. All *p* values obtained using post hoc analyses were for adjusted according to the Benjamini–Hochberg procedure.

Linear mixed effect models were used to assess the following changes:


Changes in health-related parameters between intervention groups


This concerns the anthropometrics, physical activity and overnight fasting blood parameters. In this model, ‘time’ (V2 compared with V1 and V3), ‘goal’ (yes or no) and their interaction, as well as ‘gender’ and ‘age’ were specified as fixed effects and ‘participant’ as random factor.


Changes in health-related parameters with presence of goal-setting:


To assess if setting goals for a specific food category influenced health parameters, a model was specified with ‘time’ and ‘goal’ (yes or no) and their interaction, as well as ‘gender’ and ‘age’ were as fixed effects and ‘participant’ as random factors. This hypothesis was tested within the PNA group only.

For all linear mixed models, compatibility with mixed-model assumptions was checked by inspection of residual plots and Q-Q plots. In the case of heteroscedastic residuals, data were log-transformed. Tukey (1) or Benjamini–Hochberg (2) procedures were applied when performing post hoc analyses to further identify differences within treatments as well as between time points. For these linear mixed models, statistical outliers were defined as any observation which has an absolute residual exceeding 3 times the residual standard deviation. These observations were excluded from the final models. *p* < 0.05 was considered significant in all analyses.

## 3. Results

### 3.1. Study Logistics

A total of 216 healthy males and females were recruited. Of these participants, a total of 162 subjects were found to be eligible and were randomly allocated to either the control, GNA or PNA groups (Figure 2). Baseline characteristics of the total study population (*n* = 155) and baseline characteristics per intervention group used for analysis are summarized in Table 1. No significant differences between intervention groups were observed at baseline.

### 3.2. Effect of PNA on Nutrition Intake Status and Health Compared to Control and GNA

To assess the effect of PNA on food-intake behavior, we determined changes in NIS of the PNA group between V2 and V3 and compared these to changes in the control and GNA group. The change in NIS was significantly different between the PNA group and the control and/or the GNA group (Figure 3, Appendix A). The PNA group improved NIS for fruits (*p* < 0.0001), whole grain (*p* = 0.008), unsalted nuts (*p* < 0.0001), fish (*p* = 0.0003), SSB (*p* = 0.005), added salt (*p* = 0.003) and unhealthy choices (*p* = 0.002) (Appendix A). Especially the change in number of subjects that met the dietary guidelines for fruits (from 24% to 69%; absolute: 13 to 38 subjects), unsalted nuts (from 8% to 67%; absolute: 4 to 34 subjects), fish (from 29% to 57%; absolute: 15 to 29 subjects), SSB (from 43% to 69%; absolute: 22 to 35 subjects) and making less unhealthy choices (from 35% to 51%, 18 to 25 subjects) between V2 and V3 was remarkable. Participants maintained their habitual dietary and physical activity behavior between V1 and V2, except for red meat consumption. The total study population improved their NIS for red meat significantly (from 88% to 96%; *p* = 0.04 Appendix A). When looking at between group differences, at V3, participants receiving PNA significantly improved their NIS as compared to the control and GNA group for unsalted nuts (*p* = 0.0008 PNA–control; *p* = 0.0007 PNA–GNA), fish (*p* = 0.003 PNA–control; *p* = 0.03 PNA–GNA) and added salt (*p* = 0.0008 PNA–control; *p* = 0.02 PNA–GNA), respectively.

To further determine the effect of PNA, we investigated within group changes in NIS during the intervention period (V2–V3) as compared to their run-in period (V1–V2) for the PNA group only. The change in NIS for the PNA group was significantly different for whole grains, unsalted nuts, red meat, cold cuts (meat), SSB and added salt (Appendix A). The post hoc analysis revealed a significant improvement in NIS for most of these categories, except for red meat and whole grains, during the intervention period. Red meat increased from 84 to 94% of the participants having an optimal status in the run-in period and increased from 94 to 96% in the intervention period, a change which was not significant within both periods. Whole grain consumption showed a large number of misreports at V3. These misreports probably are responsible for the difference found, while the change to optimal only increased by 2% (Figure 3C).

Subsequently, it was examined if the change in nutritional intake improved health, however none of the biologic parameters included in the study were changed due to the PNA intervention (Table 2).

### 3.3. The Effect of Goal-Setting on Nutrition Intake Status and Health Parameters in the PNA Group

Thirty-nine participants set goals for three food categories, ten participants set goals for two food categories, and one participant set goals for one food category. During the intervention period, the NIS was improved significantly in the participants who set a goal for a specific food group as compared to participants in the PNA group who did not set a goal for this specific food group (Appendix A). The NIS improved significantly for unsalted nuts (*p* < 0.0001), fruits (*p* = 0.0001), whole grain (*p* = 0.005), fish (*p* = 0.0001), dairy (*p* = 0.007), vegetables (*p* = 0.01) and unhealthy choices (*p* = 0.02) in the participants who set a goal (Figure 4). Especially the improvement in optimal status for unsalted nuts (from 0 to 28 subjects), fruits (from 0 to 18 subjects), fish (from 0 to 12 subjects), vegetables (from 0 to 4 subjects) and unhealthy choices (from 0 to 3 subjects) stands out. In contrast, the NIS deteriorates significantly for vegetables (*p* = 0.01, *n* = 42 participants did not set a goal) and dairy (*p* < 0.0001, *n* = 41 subjects did not set a goal) in the participants who did not set a goal for these food categories Appendix A. Overall, of the 120 goals which were set by the participants (*n* = 50) receiving PNA, only 8 goals were not met at the end of the intervention. It was also investigated if goal-setting improved the health status of participants. The change in biological parameters of subjects who set a goal for the three most selected food categories (unsalted nuts, fruits and whole grain) was compared with the change in biologic parameters in non-goal-setters of the PNA group for these food categories. Interestingly, significant interaction effects between time and specific goals were found for LDL cholesterol (Figure 5). Participants which set a goal for whole grains (18 of 50 subjects (36%)) decreased their LDL cholesterol to a higher degree (3.11 to 2.90 mmol/L; *p* < 0.0001) than non-goal-setters (32 of 50 subjects (64%), 3.02 to 2.92 mmol/L; *p* = 0.0003) of the PNA group. Participants who have set a goal for unsalted nuts decreased their LDL cholesterol (29 of 50 subjects (58%); 3.08 to 2.84 mmol/L; *p* < 0.0001), while non-goal-setters did not (21 of 50 subjects (42%), 3.02 to 3.00 mmol/L; *p* = 0.56).

## 4. Discussion

In the current study, we evaluated whether PNA is effective in improving food-intake behavior in a healthy population. We studied this by monitoring the change in NIS of the PNA group compared to the GNA group and control. Furthermore, we investigated if changes in food-intake behavior could improve health outcomes. In addition, for the PNA group we investigated whether setting goals for specific food groups is an effective method for improving individual food-intake behavior and if this consequently affects health. The main findings of our study were that PNA was more effective in improving NIS than generic, one-size-fits-all, dietary advice. In addition, the setting of goals for food group improvement, with the use of implementation intentions, showed to be extra beneficial in changing individual food-intake behavior.

During the free living condition, the total study population improved their NIS of red meat by 8%. This change in food-intake behavior may be explained by the fact that individuals who take part in lifestyle interventions are more likely to be motivated to adopt a healthy lifestyle compared to other individuals [37]. At V2, 96% of the participants were compliant with the nutrition guidelines for red meat [38,39]. Due to this high percentage of participants with an optimal intake of red meat, almost no improvement was possible in the intervention period. Although participants were aware that they were participating in a nutritional study no significant changes were found in NIS in the control between V1 and V2, nor between V2 and V3 indicating proper study compliance. At V2, NIS between the control and PNA group was similar.

Our findings are in line with the results from recent studies evaluating the effect of GNA and PNA on food-intake behavior [14,26]. PNA was more effective in improving food-intake behavior than GNA. Although participants knew it was a nutritional intervention it still remained difficult to change their food-intake behavior with only generic advice such as provided by public health organizations. The PNA was effective in improving the NIS of fruits, unsalted nuts, fish, SSB, added salt and unhealthy choices compared to GNA. These results suggest that personalized advice including goal-setting is more effective in changing dietary behavior than generic advice. This can be explained by the fact that personalized advice is considered more relevant and important to an individual than general advice [11,23]. For the food categories vegetables, dairy and spreads and oils, no or only a small improvement was observed in the PNA group than their run-in period or the GNA group. These results suggest that it is less attractive for participants to choose to add more vegetables to their diet, to have an optimal dairy intake or to change the type of fat they use for cooking. This can be explained by the fact that people can only cope with a few behavior changes at a time [40,41]. However, also other factors such as individual physiological or psychological traits could affect food-intake behavior [42]. For future studies, focusing on the increase of vegetables intake is advised, since meta-analyses indicate that an increased vegetable intake is associated with reduced risk of chronic disease development [43,44,45]. To improve this intake, other behavior change techniques could be incorporated. For example, problem solving, and social comparison are particularly relevant behavior change strategies for improving compliance with advice on fruit and vegetable intake [46,47].

For whole grain, the number of misreports (11 subjects) increased in the PNA group during the intervention period, since many subjects did not fill out the question on which type (white or whole grain) of bread they ate, resulting in a misreport. Figure 3C did show changes in whole grains intake, i.e., a reduction in the low whole grains intake category and an increase in both the average and optimal whole grains intake category, which could explain the observed difference. However, interpreting if the total food category improved or deteriorated was not possible. In future studies with these kind of online FFQs it would be advised to build in an answer check to prevent misreporting.

Participants in the PNA group were encouraged to set goals, by formulating implementation intentions, to optimize their NIS. Previous research already indicated that goal-setting can be an effective method for changing dietary behavior [24,47]. In contrast to a recent study [27], which used personalized advice and implementation intentions, we observed an improved dietary intake for all food categories that were chosen as goal. As this was not the case for people who did not set a goal, these results indicate that participants who have set a goal for a specific food category are more prone to improve their food-intake behavior for that food category. In the PNA group, the largest improvements were observed for the food categories unsalted nuts and fruits. These food categories were also the food categories most the participants set goals for (*n* = 29 and 23, respectively out of 50). Interestingly, for the food categories SSB and unhealthy choices, for which no or only a few participants set goals, there was also an improvement in NIS in the PNA group compared to the GNA group and to their run-in period. These results suggest that PNA is also effective in changing an individual’s behavior without setting goals.

Although the PNA group clearly improved their nutritional behavior, this did not result in a general health improvement for this group. However, when zooming in on the participants who set a goal for unsalted nuts and whole grain, we did observe changes in health than participants who did not set a goal for these food categories within the PNA group. For participants that set goals for unsalted nuts and whole grain, we observed a significantly larger decrease in LDL cholesterol than participants who did not set a goal for these food categories. These results are in line with previous findings, which showed a decrease in LDL cholesterol as a result of an increased consumption of unsalted nuts or whole grains intake [48,49,50,51].

A limitation of our study is the large variety of the subjects and their good health, making it difficult to observe health improvements. As a follow-up, it could be studied if in subjects of a specific age range with a compromised health (e.g., old age and/or overweight) a change in health parameters can be observed as a result of improved dietary intake. Second, personalized advice differs between individuals, since each individual receives their own tailored advice, increasing the difficulty of health effect analysis. Future development of personalized data analysis, even if only basic measures are present, may provide a solution for this problem such as performed by Ritz and colleagues on fasting glucose and insulin [52]. Alternatively, Zeevi et al. showed benefits of personalization when taking postprandial glucose metabolism into account [25]. Both strategies probably lack information on metabolic regulation as only parts are considered. A more complex measurement and analysis of postprandial metabolism such as performed in a recent study, in volunteers with a compromised health state who substituted refined wheat with whole grains wheat, where subtle improvements of systemic responses were observed may provide better personalized nutrition solutions [53].

A strength of our study was the use of an online web portal with automatically generated PNA, in a healthy population, thereby providing evidence that personalization can significantly improve compliance with dietary guidelines than GNA [15]. Although we did not find interaction effects on health parameters on a group level, these results may still indicate the added value of using PNA in public health initiatives. A second strength of the study was that we showed the dependency of health effects on goal-setting after nine weeks. These results may cautiously indicate that even within such a very healthy and diverse study population specific and personal health benefits could be identified.

## 5. Conclusions

In conclusion, after a nine-week intervention, healthy male and female participants receiving PNA showed an improvement of their NIS compared with participants receiving GNA. When an implementation intention was set next to receiving PNA, nutritional behavior was more prone to change. Participants who had set a goal for unsalted nuts and/or whole grains improved LDL cholesterol after nine weeks of intervention, indicating possible health benefits of PNA including goal-setting. Ongoing and future personalized nutrition studies should include postprandial metabolism to look more systemically at health of a subject. Furthermore, sustainability of the personalization strategies on health status is of interest since general (dietary) behavior change is known to be difficult to maintain.

## Figures and Tables

**Figure 1 nutrients-12-02819-f001:**
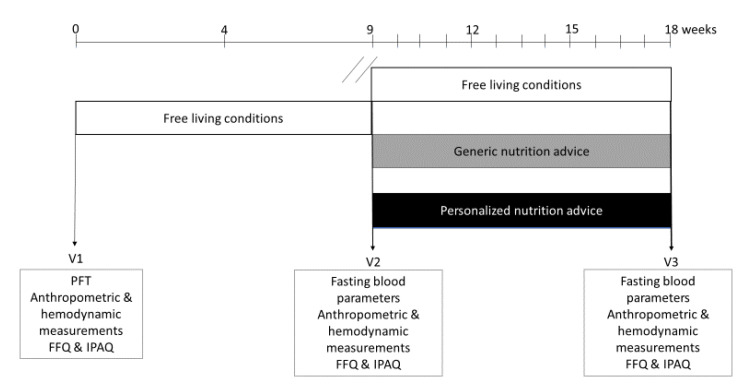
Study design. V1, V2, V3 represent the visit day numbers that the participants had their test day in the metabolic ward (Biofortis, France). Between Visits 1 and 2, the free-living period was at least nine weeks. All participants started their treatment after Visit 2 for nine weeks. Visit 3 was the last test day. FFQ—food frequency questionnaire; IPAQ—international physical activity questionnaire; PFT—PhenFlex challenge test.

**Figure 2 nutrients-12-02819-f002:**
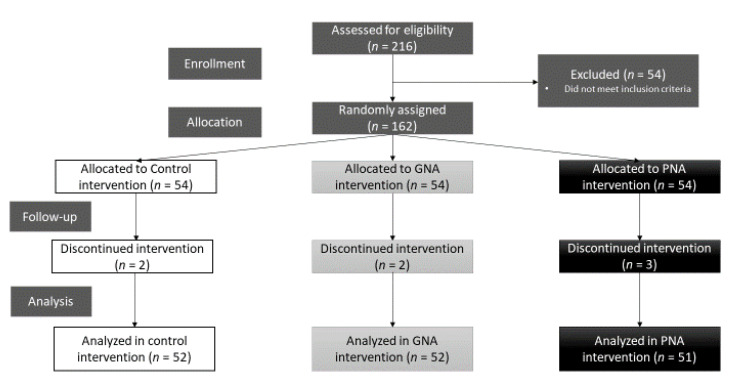
Flowchart of participation. In the control and GNA group, two participants of each group were lost to follow-up and in the PNA group, three participants were lost to follow-up. These participants stopped prematurely with the study or started using medication which could interfere with the study outcomes. GNA—generic nutrition advice; PNA—personalized nutrition advice.

**Figure 3 nutrients-12-02819-f003:**
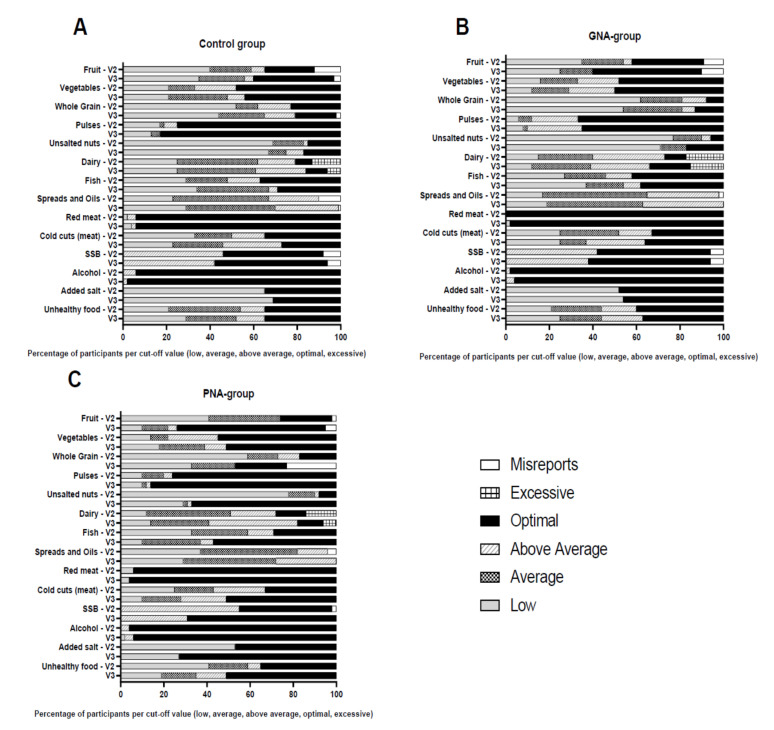
Nutrition intake status (NIS) before (V2) and after the nine-week (V3) intervention period in participants receiving (**A**) no nutrition advice (control); (**B**) GNA; (**C**) PNA. Data are percentages of participants in the control; GNA and PNA groups have a low, average, above average, optimal or excessive NIS of the 14 food categories. In white the percentage of misreports are indicated. Control: *n* = 52, GNA: *n* = 52, PNA: *n* = 51. GNA—generic nutrition advice; NIS—nutrition intake status; PNA—personalized nutrition advice; SSB—sugar-sweetened beverages.

**Figure 4 nutrients-12-02819-f004:**
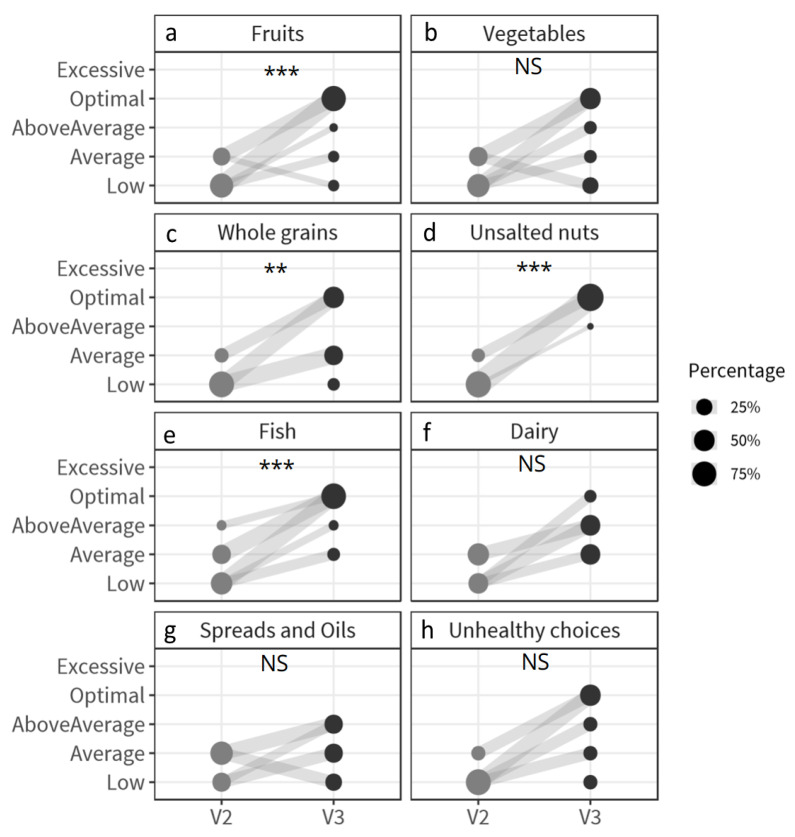
Fluctuation plots of the percentage of goal-setting participants receiving PNA per nutrition intake status before (V2) and after the nine-week intervention period (V3) for (**a**) fruit: *n* = 23; (**b**) vegetables: *n* = 8; (**c**) whole grain: *n* = 18; (**d**) unsalted nuts: *n* = 29; (**e**) fish: *n* = 16; (**f**) spreads and oils: *n* = 11; (**g**) dairy: *n* = 9; (**h**) unhealthy choices: *n* = 6. The size of the circles represents the percentage of participants who set a goal for that specific food category and the lines represent participant migration of NIS between V2 and V3. The size of the lines represent the percentage of participants that migrated between NIS during the intervention period (V2 to V3). ** *p* < 0.01, *** *p* < 0.001, NS: non-significant. NIS—nutrition intake status; PNA—personalized nutrition advice.

**Figure 5 nutrients-12-02819-f005:**
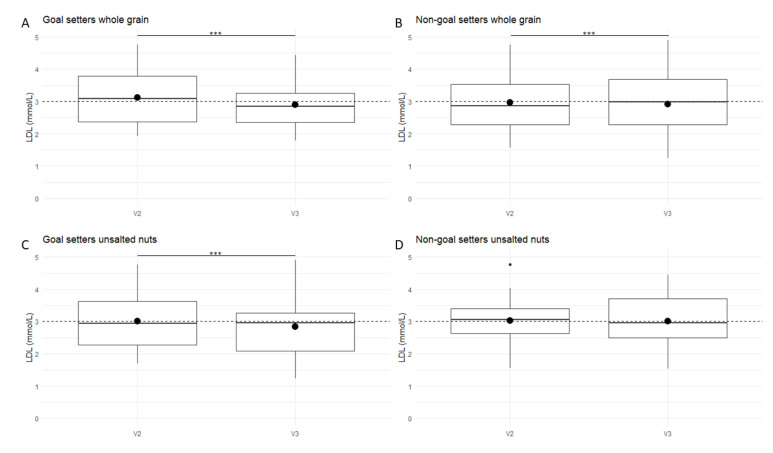
Changes in LDL cholesterol (**A**) goal-setters for whole grain, *n* = 18; (**B**) non-goal-setters for whole grain, *n* = 32 (**C**) goal-setters for unsalted nuts, *n* = 29; (**D**) non-goal-setters for unsalted nuts, *n* = 21. Data presented as descriptive boxplot, in which the center represents the median, the outlines of the box indicate quartile 1 or 25th percentile and quartile 3 or 75th percentile and the top of the line shows the most extreme data points within 1.5 times the IQR. The full circle in the boxplots represents mean LDL cholesterol. Dotted line represents an LDL cholesterol of 3.0 mmol/L. An LDL below 3.0 mmol/L is healthy, according to the Dutch Heart Association (https://www.hartstichting.nl/). *** *p* < 0.001. PNA—personalized nutrition advice.

**Table 1 nutrients-12-02819-t001:** Baseline characteristics of the study population.

Variables	Total Population	Control	GNA	PNA
Total (n)	155	52	52	51
Sex, female, *n* (%)	100 (64.5)	34 (65.4)	34 (65.4)	32 (62.7)
Age (years)	42.3 ± 12.1	42.8 ± 11.8	41.8 ± 12.1	42.3 ± 12.6
**Anthropometrics**				
BMI (kg/m^2^)	25.2 ± 3.6	25.1 ± 3.9	25.4 ± 3.7	25.1 ± 3.0
Waist circumference (cm)	83.4 ± 10.6	83.6 ± 11.3	83.5 ± 10.7	83.2 ± 10.0
Hip circumference (cm)	100.4 ± 7.8	100.2 ± 9.4	100.7 ± 7.8	100.3 ± 5.7
**Hemodynamics**				
Diastolic BP (mmHg)	73.1 ± 9.2	72.6 ± 9.6	73.9 ± 9.2	72.7 ± 8.8
Systolic BP (mmHg)	117.1 ± 13.0	116.7 ± 12.4	119.2 ± 14.4	115.6 ± 12.2
**Metabolic health fasting**				
Total cholesterol (mmol/L)	4.73 ± 0.99	4.64 ±1.05	4.90 ± 0.87	4.65 ± 1.04
HDL cholesterol (mmol/L)	1.48 ± 0.36	1.52 ± 0.39	1.50 ± 0.36	1.41 ± 0.34
LDL cholesterol (mmol/L)	2.81 ± 0.84	2.69 ± 0.81	2.96 ± 0.81	2.78 ± 0.89
Triglycerides (mmol/L)	0.29 ± 0.09	0.29 ± 0.09	0.29 ± 0.09	0.29 ± 0.1
HbA1c (%)	5.20 ± 0.24	5.21 ± 0.25	5.16 ± 0.25	5.22 ± 0.23
NEFA (mmol/L)	0.41 ± 0.19	0.41 ± 0.19	0.43 ± 0.21	0.37 ± 0.13
**Glucose and Insulin-related indices**				
HOMA-IR	1.58 ± 1.09	1.52 ± 1.14	1.61 ± 1.05	1.61 ± 1.10
**Health space analysis**				
Metabolic age	41.47 ± 11.49	41.29 ± 10.90	41.62 ± 10.48	41.51 ± 13.15

Data presented as means ± SDs. BP—blood pressure; GNA—generic nutrition advice; PNA—personalized nutrition advice; HDL—high density lipoprotein; LDL—low density lipoprotein; HOMA-IR—homeostatic model assessment of insulin resistance; BMI—Body Mass Index; HbA1c—glycated haemoglobin; NEFA—non-esterified fatty acids.

**Table 2 nutrients-12-02819-t002:** Anthropometrics and overnight fasting blood parameters before (V2) and after (V3) the intervention period in participants receiving no nutrition advice (control), GNA or PNA.

	Control		GNA		PNA		Interaction Effect	Interaction Effect
Variables	V2	V3	V2	V3	V2	V3	PNA vs. control × Time	PNA vs. GNA × Time
**Anthropometrics**								
BMI (kg/m^2)^	25.2 ± 4.0	25.4 ± 4.1	25.3 ± 3.7	25.3 ± 3.6	25.1 ± 3.0	25.1 ± 3.1	0.22	0.85
Waist (cm)	83.2 ± 11.1	83.0 ± 11.4	83.4 ± 10.8	82.6 ± 10.4	83.3 ± 10.1	82.5 ± 9.9	0.18	0.93
Hip (cm)	100.1 ± 9.4	99.6 ± 9.6	100.68 ± 8.0	99.8 ± 7.8	100.5 ± 5.7	99.8 ± 5.9	0.56	0.52
**Physical activity**								
Metabolic equivalent of task (MET) per day	732 ± 598	801 ± 777	1113 ±1005	972 ± 888	859 ± 609	842 ± 19	0.49	0.52
**Hemodynamics**								
Diastolic BP (mmHg)	73.0 ± 9.4	69.4 ± 10.4	73.9 ± 9.1	71.3 ± 10.3	72.7 ± 9.2	70.6 ± 9.0	0.35	0.72
Systolic BP (mmHg)	117.5 ± 13.1	119.5 ± 16.3	117.1 ± 14.8	121.3 ± 16.3	116.8 ± 1.7	120.9 ± 2.4	0.38	0.97
**Metabolic health fasting**								
Total cholesterol (mmol/L)	4.89 ± 1.13	4.84 ± 1.05	5.31 ± 1.09	5.19 ± 1.02	5.03 ± 1.07	4.81 ± 1.01	0.10	0.32
HDL cholesterol (mmol/L)	1.52 ± 0.37	1.52 ± 0.37	1.51 ± 0.35	1.54 ± 0.38	1.44 ± 0.34	1.41 ± 0.33	0.31	0.06
LDL cholesterol (mmol/L)	2.90 ± 0.92	2.86 ± 0.83	3.29 ± 0.96	3.18 ± 0.92	3.09 ± 0.89	2.95 ± 0.88	0.23	0.65
Triglycerides (mmol/L)	0.29 ± 0.11	0.29 ± 0.10	0.31 ± 0.11	0.3 ± 0.08	0.3 ± 0.11	0.29 ± 0.09	0.38	0.85
HbA1c (%)	5.27 ± 0.26	5.11 ± 0.28	5.25 ± 0.27	5.06 ± 0.25	5.25 ± 0.25	5.14 ± 0.24	0.39	0.10
**Glucose and Insulin-related indices**								
HOMA-IR	1.82 ± 1.07	1.88 ± 1.18	2.03 ± 2.13	1.90 ± 1.83	1.91 ± 1.61	2.04 ± 1.57	0.85	0.40

Data presented as mean ± SDs. Interaction effect represents *p* values between time and treatment where a *p* < 0.05 was considered significant; BP—blood pressure; GNA—generic nutrition advice—PNA—personalized nutrition advice; HDL—high density lipoprotein; LDL—low density lipoprotein; HOMA-IR—homeostatic model assessment of insulin resistance; BMI—Body Mass Index; HbA1c—glycated haemoglobin

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
