# Peer review of "Evaluation of Food-Intake Behavior in a Healthy Population: Personalized vs. One-Size-Fits-All"

_nutrients, 2020, doi:10.3390/nu12092819_

Round 1
Reviewer 1 Report
This interesting manuscript describes a well-designed study into the effect of personalised nutrition advice to improve dietary behaviour. Even though the long-term effect of the advice and goal setting cannot be shown in this study, the effect can be shown within the intervention and the study provides valuable insights into how the personalisation of such advice can help consumers to improve nutrition behaviour. Overall, the paper is well-written and clearly presents its main findings.
The main issue in this paper (and the study set-up) is however something that the authors can provide more insights upon: why were goals only set in the PNA group, and to what extent can the conclusion (line 330) be based on merely the PNA versus the fact that part of this group had been setting goals?
Specifically, the authors could clarify this further in these sections:
- Line 64-70: The authors seem to imply here that the study combined two things: the effects of personalized advice and the effects of goal setting. This section seems to suggest that the PNA group had both, whereas the other groups did not have both. Is this correct? Please clarify.
- Line 185: Did only PNA participants receive the advice to set a nutrition goal (which seems to be suggested in the methods section and throughout the results)? If that is the case, it can be questionable whether it is attributable to the PNA, the goal setting, or the combination of interventions that has been effective in improving health of participants. Please clarify, put forward clearly in the results and in the discussion.
- Only in section 3.3, it becomes clear that the PNA group was subdivided into two groups (goal setters versus non-goal setters). Where was the size of these groups based upon? And why were there no goals set in the GNA or no advice-groups?
- Line 330: Can the authors really contribute their main finding to the PNA versus the goal setting, or is the other way around?
Next to this issue, there are merely two minor elements that would benefit from further clarification by the authors:
- In the abstract, the authors list ‘unhealthy choices’ in the same row as improvements in specific food groups. This is slightly strange for a reader. Please rephrase the sentence in line 18-24.
- Line 180-183: The evaluation focussed on the compliance of the diet with French and Dutch dietary guidelines. Were there any differences between these guidelines and if so, how did the authors address this?
Reviewer 2 Report
Major comments:
- The aims/objectives need to be much clearer at the end of the introduction; the statistical analyses should clearly line up with them, followed by the same ordering in the results. I did not understand why each model was run until I got to the results section. Thus, consider:
- Including clearer aims: The overall aim is currently at the end of the paragraph instead of toward the beginning, which is confusing given the information in the prior sentences. Additionally, it is not quite clear what the objectives are. What are the main aims, and what are you exploring post hoc?
- Much of what should be in the statistical analyses section is currently in the results section. Why each model was run (e.g., lines 238 – 239) and decisions about those models (e.g., lines 284-285) should be pulled from the results and used to clarify the models presented in the statistical analysis section.
- The statistical analysis section needs to be reworked. The incomplete sentences and numberings are hard to follow. E.g., I’m not quite sure what lines 217 – 220 are saying. I again suggest that the aims be stated clearer at the end of the introduction, then this section better detailed so that readers can understand what was done to address each aim; what models were a priori vs. post hoc analyses.
- Once these changes are made, confirm the results are presented in the same order as the models.
- Clarify why a control vs. GNA group is needed. Are these GNA guidelines widely accessed in France (and thus could we expect the control to have access to this information as well), or are they not and thus a GNA group is needed?
- The authors mention the Benjamini-Hochberg procedure was used, but there needs to be more detail in the results. What was the false discovery rate (FDR) set to? How many tests were performed using the Benjamini-Hochberg procedure, and how many were judged to be statistically significant? This is currently unclear. Furthermore, in Figure 4, p<0.05 is indicated as statistically significant, and I question if this should be the case if this procedure was used for the goal-setting post-hoc analyses.
- You say an N of 55 is needed per group according to your power calculations, but did not reach that sample size per group. Could this limitation explain why PNA (and the control group) may not have seen a significant change within their groups, or why we do not see significant changes in blood parameters? I recommend addressing this in the Discussion limitations section or justifying the smaller N per group.
- Reconsideration should be given to what is included in the tables and text concerning change values vs. p-values. In some instances (e.g., in section 3.2), percent changes are shared, but this is not consistent. As a nutrition researcher, I am interested in seeing how large of a difference these significant changes are.
- Please submit the “data not shown” referred to for review (Lines 247 and 250), and consider including them as supplemental materials.
- The figures need to be larger to increase legibility. E.g., in Figure 3, I cannot tell the misreports from above average. In Figure 4, it is hard to tell how the legend size values line up with the figure circles. In figure 5, what are the values? Numerical labels would make this easier to understand.
- The Discussion paragraphs contain very repetitive thoughts and should be condensed. Areas of expansion to detail more on include:
- Lines 352 – 355: Could this also be due to the fact that some behaviors are harder to change, not just that one can only cope with a few changes at a time.
- Lines 362 – 363: This needs to be reworked, it is unclear what the authors are trying to say here.
- Lines 391 – 393: This study seems an odd comparison to the current study, given they looked at influencing one food group versus this study looked at changing overall diet. Are there other studies similar to the current study that can be discussed here? If not, discuss this lack of research.
- Conclusions: Future directions need to be added here, rather than just restating the results again.
- There are many old citations in this paper. E.g., line 31 ref 1 – the authors state that rates are rising, but this is a reference from 2013. Reference 6 is from 2005 (15 y old)and ref 10 is from 1999 (>20 years old) as well. In instances like ref 1, a citation within the past 5 years is more appropriate; in other cases, citations within the past 10 years is preferred. (Please note: I do see that some older references are included for specific reasons, but the number of old references is quite high without an explanation. E.g., was there a paucity of research in this specific area over the past decade?)
Minor comments:
- Abstract:
- Define NIS
- As previously mentioned, consider if more than p-values can be presented – how significant the difference/change was within/between groups is unclear.
- The future directions should be added to the abstract – the ending is currently repetitive of the results.
- I recommend a grammar check throughout. E.g., many time advices should be advice; an sometimes appears before advice but should be removed (e.g., lines 45 and 177); there are a few spaces missing and many incomplete sentences; and there are many extra commas throughout.
- Line 118 and lines 128-132: Who took the measures? Additionally, for e.g., weight measures, were they only mentioned once or in duplicate or triplicate?
- Line 154: Add (IPAQ) after Questionnaire so that it’s clear what IPAQ stands for in the next sentence.
- Line 159: Though the categories are provided in a supplemental table, there should be a citation for the leaflet for readers to access both in the text and in the table footnote.
- Line 164: remove questionnaire (repetitive of the Q in FFQ)
- Line 230: It looks like the footnote is currently merged with the figure title. Please separate them.
- There is data presented from Supplemental Table 3 but it is not called out on line 243.
- Line 248: You mention physical activity was maintained across groups, but this data is currently missing. I recommend including baseline mean MVPA (or the PA measure examined at V1 – V3) in Table 1.
- Lines 252 – 253: the “or” between p-values is confusing. I suggest reworking this sentence.
- Line 337: You call out Supplemental Table 3, but this data does not seem to appear there.
Reviewer 3 Report
Abstract
line 22: Use of NIS is out of the blue because this abbreviation was not previously defined. I believe it means 'nutritional intake status'. If so then put NIS in parenthesis after the full definition on line 19.
Line 21-24: Sentence 'Goal setting led to...observed' does not read well. The sentence is unclear. Maybe something along the lines of 'goal setting led to improvements in NIS, including....
Methods
line 79: Replace 'which' with 'who'
line 83: It will be clearer and more stand-alone if the sentence is revised to read as …(TFEQ, restraint scores of < 13 were included) rather than ...factor 1<13 were included.
line 100: Put a space between the period after the parenthesis and the beginning of the next sentence.
line 107: Formatting of the text is off
Results
line 222: Title for section 3.1 is presented as one word. Should it perhaps be 'study logistics' ?
Table 1: Format the row one headings to be in vertical alignment with the values in the subsequent rows under those headings. The headings in the first row seem to be center aligned whiles the values in the table seem left justified. Consider making the values centered as well for alignment.
Also in Table 1, check the closing bracket for the units for BMI, this should not be a superscript.
Consider making figure 3 larger than it is now for legibility. There seems to be enough white paper to accommodate a larger version of that figure. It is hard to read clearly.
Discussion
Line 387: Change 'comprised' to 'compromised'
Based on your results, it appears you did not observe a significant improvement in NIS between the control group and the PNA group from V2 to V3. Can you speculate in your discussion why this was the case? Would it be fair to say that free-living approach has the same effect as PNA in terms of effects on changes in NIS?
Round 2
Reviewer 2 Report
Thank you for your great edits - this revised paper is significantly stronger than the previous version with better flow, increasing readability and drawing the reader in to a really nice study. The suggested edits below are minor, but I believe will strengthen the paper further.
Abstract, Line 21: Consider rewarding to "Coal-setting led to greater improvement," as the usage of the word prone was confusing.
Introduction:
- Line 37: Authors cite the US DGA but call out both the US and Europe. I just wanted to check and see if there is a citation(s) that can be added for Europe, too.
- Lines 65 and 99: Replace personalized nutrition advice with PNA here since it is already defined on line 38.
- Line 67: Remove "Therefore" so it is not in two consecutive sentences for flow.
- Line 21 vs. 69 vs. 70: Authors switch back and forth between goal-setting, goalsetting, and goal setting throughout the paper. I suggest using one version throughout for consistency.
Methods:
- Double-check for extra commas throughout still (E.g., after kg/m2 on line 82, after intolerances in line 89, after participants on line 212, after although on line 394, etc.
- Line 159: Pluralize lifestyles
- Lines 168 and 180: Double-check sign usage. Where do men = 20% and women = 30% fall, and where do those =75% fall, respectively? Make sure they are included on line 168 and there is no overlap in categories on line 180.
- Lines 213 - 215, 221-223, and 225-227: The statistical analysis section is vastly improved, thank you! However, I do suggest reworking these lines to make sure they are clear - I currently do not understand them fully.
- I appreciate the authors explained who took the measures (previous minor comment for line 118 and 128-132). Please include this information about trained personnel and measures taken only once in the methods section
Supplemental Table 1: For the recommendation for milk and dairy products there is a semi-colon after salty - is this in error or is there text missing?
Supplemental Table 3: Remove duplicate definition of PNA in the footnote
Results:
- Table 1: Define HDL, LDL, and HOMA-IR in the footnotes
- Table 2 and supplemental tables 1 - 2: Add the p-value for significance in the footnotes
- Lines 257-259 and 300-301: I believe the authors may have misinterpreted my comment #5. Percent change is ideal, in these sections over absolute values, I just meant that percent change would be useful to include if possible in other areas where significant changes occurred within groups.
- Figure 4: Where are the p-values in the footnotes indicated in the Figure? Which are significant changes?
Discussion:
- Line 338: Change control group to control (since it's referred to as control throughout)
- Line 348: This should be participants were compliant
- Line 271: Remove" which" after social comparison to make this a complete sentence
- Lines 377-378: Consider rewording this or grammar and flow
- Line 379: Change food frequency questionnaires to FFQs since the abbreviation is used earlier
